# Prevalence and virulence gene profiles of *Escherichia coli* O157 from cattle slaughtered in Buea, Cameroon

**Elvis Achondou Akomoneh**[1]*, **Seraphine Nkie Esemu**[2,3], **Achah Jerome Kfusi**[2,3], **Roland N. Ndip**[2,3], **Lucy M. Ndip**[2,3,4]

1 Department of Biological Science, Faculty of Science, University of Bamenda, Bamenda, Cameroon,
2 Department of Microbiology and Parasitology, Faculty of Science, University of Buea, Buea, Cameroon,
3 Laboratory for Emerging Infectious Diseases, University of Buea, Buea, Cameroon, 4 Center for Tropical Diseases, University of Texas Medical Branch, Galveston, TX, United States of America

* eakomoneh@gmail.com

**Data Availability Statement:** All relevant data are within the manuscript and its Supporting Information files. Additionally, representative sequences; five stx2, five eaeA and nine hlyA, are accessible in GenBank under accession numbers

## Abstract

### Background

*Escherichia coli* O157 is an emerging foodborne pathogen of great public health concern. It has been associated with bloody diarrhoea, haemorrhagic colitis and haemolytic uremic syndrome in humans. Most human infections have been traced to cattle and the consumption of contaminated cattle products. In order to understand the risk associated with the consumption of cattle products, this study sought to investigate the prevalence and identify virulence genes in *E. coli* O157 from cattle in Cameroon.

### Method

A total of 512 rectal samples were obtained and analysed using conventional bacteriological methods (enrichment on modified Tryptone Soy Broth and selective plating on Cefixime-Tellurite Sorbitol Mac-Conkey Agar) for the isolation of *E. coli* O157. Presumptive *E. coli* O157 isolates were confirmed serologically using E. COLIPRO™ O157 latex agglutination test and molecularly using PCR targeting the *rfb* gene in the isolates. Characterisation of the confirmed *E. coli* O157 strains was done by amplification of *stx1*, *stx2*, *eaeA* and *hlyA* virulence genes using both singleplex and multiplex PCR.

### Results

*E. coli* O157 was detected in 56 (10.9%) of the 512 samples examined. The presence of the virulence genes *stx2*, *eaeA* and *hylA* was demonstrated in 96.4% (54/56) of the isolates and *stx1* in 40 (71.4%) of the 54. The isolates exhibited three genetic profiles (I-III) with I (*stx1*, *stx2*, *eaeA* and *hlyA*) being the most prevalent (40/56; 71.4%) while two isolates had none of the virulence genes tested.

### Conclusion

A proportion of cattle slaughtered in abattoirs in Buea are infected with pathogenic *E. coli* O157 and could be a potential source of human infections. We recommend proper animal

MG481819-MG491823 (stx2), MG458417-
MG458421 (eaeA) and MG491810-MG491818
(hlyA).

**Funding:** The authors received no specific funding
for this work.

**Competing interests:** The authors have declared
that no competing interests exist.

food processing measures and proper hygiene be prescribed and implemented to reduce
the risk of beef contamination.

## 1. Introduction

Enterohemorrhagic *Escherichia coli* serotype O157 is a well recognised human pathogen associated with bloody diarrhoea, haemolytic uremic syndrome, and death in both developing and industrialised nations [1]. Human infections with *E. coli* O157 are often attributed to the consumption of contaminated food or water with cattle at the centre of the transmission cycle. Cattle serve as the main reservoir from where the bacteria may be disseminated to humans via contaminated products or through the food chain [2, 3]. Common sources of human infections include the consumption of infected or contaminated milk, contaminated beef hamburger, vegetables, and drinking water [4] although person-to-person transmission is also possible [5].

Cattle acquire the pathogen via the faecal-oral route through consumption of contaminated feed, water, or by direct contact with the environment or other animals [6]. Infections generally do not result in disease condition and the bacteria typically colonise the lower gastrointestinal tract [6] from where they are shed intermittently in bovine faeces [7]. Shedding is by both adult cattle and weaned calves [8]. *E. coli* O157 can survive and replicate in cattle dung for more than 20 months [9]. This presents the possibility of the organism to persist in the environment and ensure maintenance and transmission.

Unlike in cattle, *E. coli* O157 induces injury in humans due to its ability to produce numerous virulence factors, most notably Shiga toxin (Stx), which is one of the most potent toxins reported so far [10]. The Shiga toxin is a phage encoded exotoxin and has two major forms; Shiga toxin 1 (Stx-1) and Shiga toxin 2 (Stx-2) [11] and acts by inhibiting protein synthesis in endothelial and other cells [12]. The major effect is damage to vascular endothelium, leading to thrombotic lesions and disseminated intravascular coagulation. This does not occur in cattle because they lack the toxin receptors in their blood vessels. In addition to toxin production, *E. coli* O157 produces numerous other putative virulence factors including a protein called intimin, which is responsible for intimate attachment of the bacteria to the intestinal epithelial cells, causing attaching and effacing lesions in the intestinal mucosa and aiding in the attachment and colonisation of the bacteria in the intestinal wall [13]. Intimin is encoded by the chromosomal gene *eaeA*, which is part of a pathogenicity island termed the locus of enterocyte effacement. Another factor that may also affect virulence of *E. coli* O157 is haemolysin (encoded by *hlyA* gene) which can lyse red blood cells and liberate iron to help support *E. coli* metabolism [14].

Despite reports of the wide geographic distribution of *E. coli* O157 in the African continent since its first reported outbreak in 1982 [15], and the role played by cattle in the transmission cycle of the pathogen, there is a paucity of information on the epidemiology of *E. coli* O157 in Cameroon. This study sought to investigate the prevalence of *E. coli* O157 in cattle slaughtered in Buea municipality and to determine the virulence profiles. Data generated in this study are necessary to create public awareness as well as inform preventive and control measures.

## 2. Materials and methods

### 2.1 Sample collection

A total of 512 rectal samples were collected from freshly slaughtered cattle at the two abattoirs (Muea abattoir and Buea town abattoir) in the Buea municipality (S1 Fig) during the period of

October 2015 to September 2016. Samples were collected from all the cattle slaughtered at each abattoir on the selected days (Mondays and Saturdays; Buea town abattoir, Tuesdays and Thursdays; Muea abattoir) during the collection period. About 40 g of faeces were collected immediately after evisceration from the bovine rectum. The rectum of each cattle was aseptically squeezed, and the content emptied into a well labelled sterile universal bottle. Samples were collected once from each cattle. The cattle slaughtered at the abattoirs are brought from all over Cameroon and the meat sold to the inhabitants of the municipality and beyond. All samples collected were transported at 4 °C, within 2 h of collection, to the Laboratory for Emerging Infectious Diseases, University of Buea for analyses.

## 2.2. Ethics approval and consent

This was an observational prevalence study performed on animals already slaughtered for consumption and therefore in the context of safe handling and quality improvement. This study was approved by the Department of Microbiology and Parasitology, University of Buea and authorised by the administrators of the Buea abattoirs.

## 2.3. Isolation and identification of presumptive *E. coli* O157 strains

Each sample was pre-enriched in modified Tryptone Soy Broth (mTSB) [16] (Tryptone Soy Broth, Oxoid CM989 supplemented with 20 mg/L of novobiocin, OxoidSR0181). Briefly, 1 mL of unformed stool or 1 g of formed stool was transferred into 9 mL mTSB, homogenised by vortexing for 5 min at 120 rpm and incubated overnight at 37°C. The pre-enriched culture (0.1 mL) was then plated onto Cefixime-Tellurite Sorbitol MacConkey (CT-SMAC) agar (sorbitol MacConkey agar, Oxoid CM981 supplemented with 2.5 mg/L cefixime and Potassium Tellurite, Oxoid, SR0172E) and incubated at 37°C for 24 h. Non-Sorbitol-Fermenting (NSF) colonies (colourless, circular and entire edge colonies with brown centres) were considered presumptive *E. coli* O157 strains whereas pinkish coloured colonies (Sorbitol-Fermenters) were considered non-O157 *E. coli* strains [17]. All NSF colonies were subcultured on Eosin Methylene Blue (EMB) agar and incubated at 37°C for 24 h. The suspected *E. coli* O157 colonies that were retained for further identification, appeared dark red to purple red with green metallic sheen on EMB agar and stained Gram negative.

## 2.4. Confirmation of presumptive *E. coli* O157 strains

All presumptive *E. coli* O157 strains were tested serologically using the *E. coli* O157 latex agglutination test E. COLIPRO™ O157 KIT (Hardy diagnostics, USA) according to manufacturer's instructions. Seropositive strains were further confirmed by the detection of *E. coli* $rfb_{EO157}$ gene using PCR. Only PCR positive strains for $rfb_{EO157}$ gene were retained for analysis. Total DNA was extracted from pelleted cells of freshly prepared broth cultures of *E. coli* O157 isolates using the QIAamp DNA Mini Kit following the manufacturer's instructions (QIAGEN, Germany). The eluted DNA was held at -20°C until used for PCR analyses. Each reaction mixture was made up of template DNA (5 μL), 12.5 μL of 2X master mix (TopTaq™ Master Mix, Qiagen, Hilden, USA) and 0.5 μL of each primer from a working solution of 20 μM (final concentration of 0.4 μM), and nuclease-free PCR water to make up 25 μL total individual reaction volume. The amplification of DNA was carried out in a GeneAmp PCR system 2700 thermal cycler (Applied Biosystems, USA). PCR targeting the $rfb_{EO157}$ gene was optimized at 95°C for 15 min; 30 cycles of 94°C for 30s, 56°C for 1 min and 72°C for 1 min; and a final extension of 72°C for 10 min. The primer set used were O157f (CGGACATCCATGTGATATGG) and O157r (TTGCCTATGTACAGCTAATCC) as previously described [18].

## 2.5. Characterisation of *E. coli* O157 strains

*Escherichia coli* O157 virulence genes for Shiga toxins 1 and 2 (*stx1* and *stx2* respectively), attaching and effacing (*eaeA*) and haemolysin (*hlyA*) were investigated. Singleplex and multiplex PCR reactions were performed and each individual reaction had a final volume of 25 μL. Singleplex *stx1* PCR was performed under the following cycling conditions: 94˚C for 5 min; 30 cycles of 94˚C for 30s, 52˚C for 1 min and 72˚C for 1 min; and a final extension of 72˚C for 10 min. Multiplex PCR targeting the *stx2*, *eaeA* and *hlyA* genes was optimized at 95˚C for 15 min, 30 cycles of 94˚C for 1 min, 56˚C for 1 min and 72˚C for 1 min; and a final extension of 72˚C for 10 min. Primer sets used for the amplification of the virulence genes [*stx1* (stx1f and stx1r), *stx2* (stx2f and stx2r), *eaeA* (eaeAf and eaeAr) and *hlyA* (hlyAf and HlyAr)] were as described previously by Paton and Paton [18]. PCR products were separated on 1.5% agarose gel stained with SYBR safe DNA Gel Stain (Invitrogen, Thermo Fisher Scientific, USA) and visualised under UV light using a Gel Documentation-XR (BIORAD, Hercules, CA).

## 3. Results

A total of 512 faecal samples (one per cattle) were collected from the two abattoirs between October 2015 and September 2016. Gross unsanitary practices were observed at the abattoirs. At both abattoirs, cattle were slaughtered on the abattoir floor which is not disinfected after every kill, and butchers walked between carcasses as they go about their daily activities. Furthermore, personnel with abrasions on fingers or hands continued to handle carcass or edible organs. Both abattoirs are constructed beside streams probably to facilitate washing and waste disposal. There was absence of tap water and washing was mostly done either in the nearby stream or in a bucket of water carried from the stream and all waste generated are channeled back through specially designed sewerage systems into the stream (Fig 1). The streams are used for irrigation, recreation and domestic purposes a few meters downstream of the abattoir (Fig 1). There was total absence of hot water, steriliser, retention room (cooling facilities), change rooms or bathroom facilities in the abattoirs.

Of the 512 samples examined, 56 showing colony characteristic of *E. coli* O157 on CT-SMAC and EMB culture media, were confirmed by latex agglutination and PCR, giving a prevalence rate of 10.9%. The presence of four virulence genes, *stx1*, *stx2*, *eaeA* and *hlyA*, was assessed in the 56 *E. coli* O157 isolates using singleplex and multiplex PCR assays. Amplified PCR products were separated on a 1.5% agarose gel (Fig 2).

Fifty-four isolates expressed at least three virulence genes while 2 isolates did not have any of the virulence genes assessed. The most prevalent virulence genes were *stx2*, *eaeA*, and *hlyA* which were each detected in 96.4% (54/56) of the isolates. The Shiga toxin 1 virulence gene (*stx1*) was detected in 40 (71.4%) of the isolates. Three virulence gene profiles (I-II) were obtained (Table 1). Forty (71.4%) isolates exhibited all the four virulence genes tested (profile I).

## 4. Discussion

Food animals are recognised sources of human *E. coli* O157 infections [19], with domesticated ruminants, especially cattle, established as major natural reservoirs for the pathogen and playing significant roles in the epidemiology [2, 3]. The consumption of undercooked meat products from cattle has been often associated with many human infections. Beef constitutes a staple food in Cameroonian cuisines and the consumption of bovine products in various forms such as beef hamburger, milk and fermented yogurt is common on Cameroonian menus [20]. Vegetables, reported as important vehicles for *E. coli* O157 transmission to humans [21], are cultivated in Cameroon with bovine faeces as manure and irrigated with

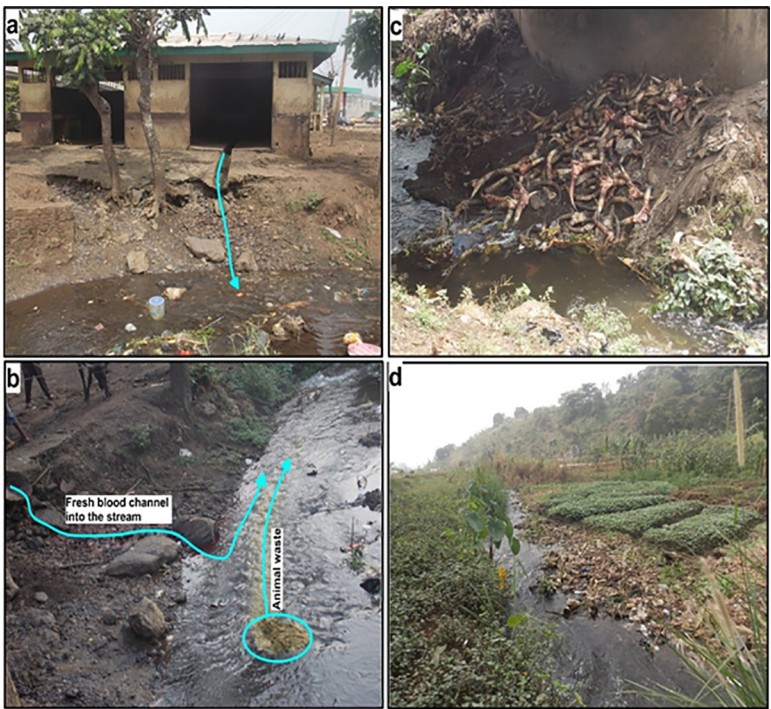

**Fig 1. Muea abattoir and wastes released into nearby stream: a, abattoir room with waste channelled to the nearby stream; b, fresh blood running into the stream and cattle intestinal content deposited in the stream; c, other waste deposited in the stream; d, vegetable cultivation downstream of abattoir.**

water sometimes contaminated with bovine faeces [22]. These vegetables are also integral in Cameroonian diet, eaten raw as in salad, coleslaw and beef hamburger as well as in other traditional dishes [23]. Our study which sought to establish the prevalence of *E. coli* O157 in cattle slaughtered in abattoirs in Buea, Southwest Region of Cameroon confirmed the presence of the highly infectious *E. coli* O157 pathogen in cattle with a prevalence rate of 10.9%. These cattle are brought into Buea abattoirs from all over Cameroon and are reared mostly by individual households for subsistence using different farming methods mostly inherited as a culture [24].

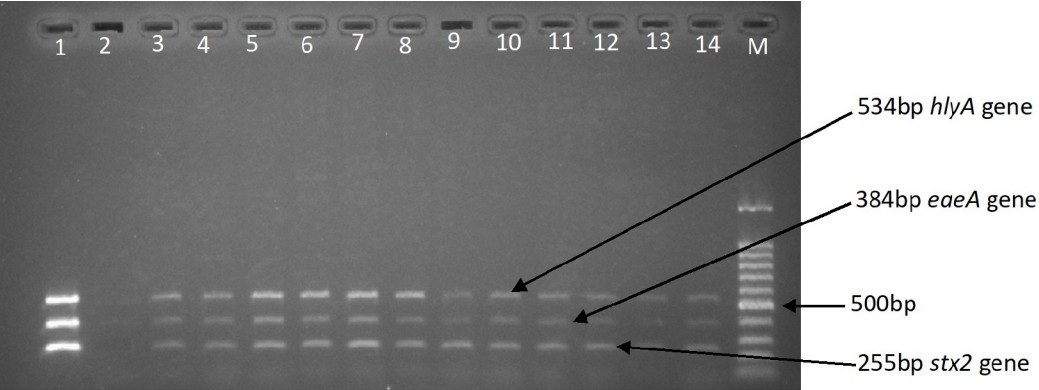

**Fig 2. Electrophoretic separation of amplified PCR products of multiplex *stx2, eaeA and hlyA* PCR.** Positive control (lane 1), negative control (lane 2), positive samples (lanes 3–14), 100 bp DNA ladder (lane M).

Table 1. Virulence profiles of *E. coli* O157 obtained in this study.

| Virulence gene profile | Shiga toxin genes | | | | Number of *E. coli* O157 isolates (%) |
|---|---|---|---|---|---|
| | *Stx1* | *Stx2* | *eaeA* | *hlyA* | |
| I | + | + | + | + | 40 (71.4) |
| II | - | + | + | + | 14 (25) |
| III | - | - | - | - | 2 (3.6) |

+, present; -, absent.

The prevalence rate reported in the present study is of great concern for the entire country. Poor sanitary practices in slaughter houses may lead to contamination of carcasses [25] which are eventually used for food [26]. Similar studies around the world have also reported *E. coli* O157 prevalence in cattle with consistent findings. Luga *et al.* [27] reported a prevalence rate of 9% in neighbouring Nigeria and Hiko *et al.*, [28] and Bekele *et al.* [29] respectively reported 8% and 10.2% in Ethiopia. Callaway *et al.* [30] reported a prevalence rate of 11.3%, in the United States, Synge *et al.* [31] and Omisakin *et al.* [32] reported prevalence rates of 8.6% and 7.5% respectively in the United Kingdom while Hashemi *et al.* [33] had a prevalence rate of 8.3% in Iran. It is worth noting that, though the prevalence rates in cattle are somewhat similar in developing and developed countries, the risk posed is not the same as most of the developed countries have intact food safety measures with intricate standards for food preparation aimed at preventing the transmission of pathogens along the food chain. These include recommendations from the Food and Agriculture Organization of the United Nations, ISO 22000 standard and Hazard Analysis and Critical Control Point (HACCP) principles. In Australia for instance, Food Standards Australia New Zealand requires basic food safety training for at least one person in each food business [34]. In the US, the Food Safety and Inspection Service (FSIS) is primarily responsible for the safety of meat, poultry, and processed egg products. In addition, FSIS is also charged with administering and enforcing the Federal Meat Inspection Act and the Humane Slaughter Act. The FSIS inspection program personnel inspect every animal before slaughter, and each carcass after slaughter to ensure public health requirements are met [35]. Such rigorous controls are lacking in most developing countries or when present are not properly implemented as most governments in these regions do not appreciate the major public health and economic implications of food safety [36]. Implementation sometimes only follows an outbreak and with individual citizens even taking the challenge. A case in point was in Kenya when a Rift Valley fever outbreak let to consumers of ruminant meat demanding to see a butcher's certificate [37]. Considering the low infectious dose (<10 cells) required for human *E. coli* O157 infection to be established, there is a pending danger of an outbreak if adequate public health and food safety measures are not put in place to contain its spread.

The pathogenicity of *E. coli* O157 is associated with a number of virulence factors, notably the cytotoxic Shiga toxins 1 and 2, the adherence intimin factor and the haemolysin factor. We also investigated the presence of four genes associated with these virulence factors in all the 56 *E. coli* O157 isolates using PCR. All four genes (profile I) were detected in 40 isolates while 14 isolates each had three of the four genes (*stx2*, *eaeA* and *hlyA*). Two of the isolates had none of the four virulence genes tested. The differences observed in the number of virulence genes tested in the isolates could arise from ecological factors as the samples analysed were collected from cattle originating from different regions in Cameroon. These findings are in line with similar studies that investigated the presence of multiple virulence genes in the same *E. coli* O157 strain. Cagney *et al.* [38] showed all 43 *E. coli* O157 isolates in minced beef and beef burgers from butcher shops and supermarkets in the Republic of Ireland contained the *eaeA* and

*hylA* genes while 41 of the isolates contained both the *stx1* and *stx2* genes. Khalid *et al.* [39] in a study in Egypt, found a combination of the four virulence genes, *stx1*, *stx2*, *eaeA* and *hlyA* in 46.7% (7/15) of *E. coli* O157 strains, while Mohammad *et al.* [40] identified *stx2* in 90.91% (10/11) of *E. coli* O157 isolates in a study that investigated the presence of multiple virulence genes in the pathogen in Tabriz, Iran. Similar findings were reported by Leotta *et al.* [41] in both New Zealand (89%) and Argentina (91%) where the prevalence of *stx2* was found to be much higher than *stx1*.

The Shiga toxin is the most important toxin associated with the pathogenicity of *E. coli* O157. The high prevalence of *stx2* (54/56; 96.6%) is of great concern considering that Stx-2 is 1000 times more potent than Stx-1 [10] and is mostly implicated in cases of haemolytic uremic syndrome in humans [42]. The *eaeA* gene detected encodes the adherence factor intimin, which is required for intimate attachment of *E. coli* O157 to the host intestinal mucosa [43] and has been identified as an important accessory virulence factor that correlates with disease [44]. The *hlyA* gene encodes another major virulence factor in *E. coli* O157 and its detection alongside the other virulence genes in the recovered strains increases the ability of the organisms to cause infection and severe illnesses [45], highlighting the danger posed by the continuous circulation of this pathogen.

It is worth nothing that, the isolation of *E. coli* O157 was on Sorbitol MacConkey agar and only non-sorbitol fermenting colonies were further analysed. This was a limitation in this study as it could have underestimated the prevalence of the pathogen.

## 5. Conclusions

Cattle serving as food animals could be carriers of highly virulent *E. coli* O157 which have potentials to cause severe infections in humans. To the best of our knowledge, this is the first study to delineate the potential of cattle as reservoirs *of E. coli* O157 in Cameroon. Considering that the circulation of this pathogen in cattle poses an environmental and human risks, we recommend that abattoir workers should be trained on the basic concepts and requirements of food and personal hygiene as well as those aspects particular to the specific food-processing operation including waste disposal. Also, municipal authorities should relocate slaughter houses away from streams in order to discourage the disposal of waste into neighbouring streams and equip the abattoirs with washing and waste disposing facilities.

## Supporting information

**S1 Fig. The map of Buea indicating the two abattoirs in the municipality.**
(TIF)

## Acknowledgments

The Laboratory for Emerging Infectious Diseases, University of Buea provided reagents, materials and equipment to accomplish this work. Our appreciation goes to the administration and workers at the abattoirs in Buea for granting us the permission to work in their establishments. We are grateful to Professor Pascal O. Bessong of the University of Venda, South Africa for assistance in sequencing the amplicons.

## Author Contributions

**Conceptualization:** Elvis Achondou Akomoneh, Roland N. Ndip, Lucy M. Ndip.

**Data curation:** Elvis Achondou Akomoneh, Seraphine Nkie Esemu, Achah Jerome Kfusi.

**Formal analysis:** Elvis Achondou Akomoneh, Seraphine Nkie Esemu, Lucy M. Ndip.

**Investigation:** Elvis Achondou Akomoneh, Lucy M. Ndip.

**Methodology:** Elvis Achondou Akomoneh, Seraphine Nkie Esemu, Achah Jerome Kfusi, Lucy M. Ndip.

**Supervision:** Roland N. Ndip, Lucy M. Ndip.

**Validation:** Lucy M. Ndip.

**Writing – original draft:** Elvis Achondou Akomoneh, Roland N. Ndip, Lucy M. Ndip.

**Writing – review & editing:** Seraphine Nkie Esemu, Roland N. Ndip, Lucy M. Ndip.

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
