## [Decision Letter · Decision Letter 0]

24 Aug 2020

PONE-D-20-18250

Prevalence and Virulence Gene Profiles of Escherichia coli O157 from Cattle Slaughtered in Buea, Cameroon

PLOS ONE

Dear Dr. Akomoneh,

Thank you for submitting your manuscript to PLOS ONE. After careful consideration, we feel that it has merit but does not fully meet PLOS ONE’s publication criteria as it currently stands. Therefore, we invite you to submit a revised version of the manuscript that addresses the points raised during the review process.

The manuscript needs revision of all sections, updating references and addressing discrepancies in results and highlighting limitations.

We look forward to receiving your revised manuscript.

Kind regards,

Iddya Karunasagar

Academic Editor

PLOS ONE

Journal Requirements:

2. Please amend the manuscript submission data (via Edit Submission) to include author Seraphine Nkie Esemu, Achah Jerome Kfusi, Roland N. Ndip and Lucy M. Ndip.

3. Your ethics statement must appear in the Methods section of your manuscript. If your ethics statement is written in any section besides the Methods, please move it to the Methods section and delete it from any other section. Please also ensure that your ethics statement is included in your manuscript, as the ethics section of your online submission will not be published alongside your manuscript.

Additional Editor Comments (if provided):

Two reviewers have commented on the manuscript a a a number of concerns have been raised. The authors need to revise the manuscript addressing all the comments point by point. The discussions should be in the light of recent references. The discrepancies in the number of colonies studied and confirmed needs to be corrected. Lack of testing of sorbitol fermenting colonies can be mentioned as a limitation of the study.

Reviewers' comments:

Reviewer's Responses to Questions

**Comments to the Author**

1. Is the manuscript technically sound, and do the data support the conclusions?

Reviewer #1: No

Reviewer #2: Yes

2. Has the statistical analysis been performed appropriately and rigorously? 

Reviewer #1: N/A

Reviewer #2: N/A

3. Have the authors made all data underlying the findings in their manuscript fully available?

Reviewer #1: Yes

Reviewer #2: Yes

4. Is the manuscript presented in an intelligible fashion and written in standard English?

Reviewer #1: No

Reviewer #2: No

5. Review Comments to the Author

Reviewer #1: PONE-D-20-18250

Prevalence and Virulence Gene Profiles of Escherichia coli O157 from Cattle Slaughtered in Buea, Cameroon

This study analyzed 512 rectal samples from abattoirs for E. coli O157 using selective enrichment and isolation, followed by confirmation by latex agglutination and rfb-specific PCR. E. coli O157 was isolated from �11% of the samples, and most of them carried stx1 and/ stx2, along with hlyA and eaeA genes. The authors conclude that the cattle slaughtered for beef could be a potential threat to beef consumers in the region of their study.

This study reports what has already been reported from all over the world, and hence lacks novelty as such. The methods followed in this study do not add any advantage in terms of providing some new information or insight into the diversity, prevalence or virulence of E. coli O157 other than what is known about them. Hence, no new knowledge is generated. This study is vastly of local interest.

A brief analysis of the study reveals why no novel or interesting information is generated from this and similar studies. First, the isolation method used for O157 is conventional and does not have any added advantage in terms of improved isolation of typical or atypical strains of O157. Isolation of O157 based on sorbitol non-fermenting phenotype alone can underestimate their prevalence. Authors should also have tested equal number of sorbitol fermenting colonies. Concentration of O157 using immunomagnetic beads can improve the isolation of O157 including atypical strains. This is important since the study aims to understand the disease burden due to O157 through the consumption of beef.

The methodology followed for isolation is not clear. How many colonies were picked from each selective plate? The manuscript says “all NSF” colonies. This confusion extends to results section too (absence of line numbers in the manuscript makes it difficult to precisely point out the lines). In page 7, last paragraph, it is mentioned that 56 samples yielded E. coli O157, and all 56 colonies were confirmed by latex agglutination test and RfbO157 PCR. That’s 100 specificity of isolation, latex agglutination and PCR assays!. It is important to explain how many colonies were presumptively selected as E. coli, and how many of these were O157 and how many were not O157. When such investigations are done, multiple strains of same serotype/genotype are isolated, and these could be clonal or non-clonal. Comparison of these would be scientifically interesting and epidemiologically valuable.

The authors say that 10% prevalence is of great concern. However, it must be noted and as have authors mentioned in multiple places in the manuscript, that Shiga toxin-producing E. coli could be naturally associated with cattle, including healthy ones. In other words, STEC are not secondary contaminants in beef unlike vegetables, fish or chicken. In this context, it is also important to know the carriage/faecal shedding rate of O157 in healthy cattle, as well as in beef. If 10% is the prevalence rate in the abattoir, what percentage of beef in markets has these bacteria? Does faecal carriage simply indicate their prevalence in beef? These pertinent questions are not answered in this investigation.

Reviewer #2: Prevalence and Virulence Gene Profiles of Escherichia coli O157 from Cattle Slaughtered in Buea, Cameroon by Akomoneh and others

The authors investigated the prevalence and identified virulence genes in E. coli O157 from slaughtered cattle in Cameroon. A total of 512 rectal samples were collected and analysed using conventional bacteriological methods followed by characterization of the confirmed E. coli O157 strains using amplification of stx1, stx2, eaeA and hlyA virulence genes by PCR. Overall this is a good study and the sample size is reasonable. The manuscript needs to be corrected for usage of language.

Major comment:

The review of literature is not recent.

There are no references cited published after 2015

Only 14 references cited are published after 2010.

The authors should narrate some suggestions for improvement of abattoir conditions in Cameroon.

Minor comments

Change Minutes to min and hours to h throughout the manuscript.

Page 6, line 2: E. coil E. coli

Page 9, line 9: Our prevalence rate of 10.9% is of great concern for the entire country

Change to : The prevalence rate reported in the present study…….

Page 9, line 14: United State United States

Line 17 : in cattle is somewhat in cattle are somewhat

Fig 2. The names of the genes should be italicized.

6. PLOS authors have the option to publish the peer review history of their article (what does this mean?). If published, this will include your full peer review and any attached files.

Reviewer #1: No

Reviewer #2: No

---

## [Author Response · Author response to Decision Letter 0]

3 Nov 2020

Dear Editor, Reviewers,

Thank you for the review comments. Responses have been provided along each review’s comments on the attached "Responses to reviewers". We hope this is adequate.

---

## [Decision Letter · Decision Letter 1]

2 Dec 2020

Prevalence and virulence gene profiles of Escherichia coli O157 from cattle slaughtered in Buea, Cameroon

PONE-D-20-18250R1

Dear Dr. Akomoneh,

We’re pleased to inform you that your manuscript has been judged scientifically suitable for publication and will be formally accepted for publication once it meets all outstanding technical requirements.

Kind regards,

Iddya Karunasagar

Academic Editor

PLOS ONE

Additional Editor Comments (optional):

The reviewer comments have been addressed satisfactorily

Reviewers' comments:

Reviewer's Responses to Questions

**Comments to the Author**

1. If the authors have adequately addressed your comments raised in a previous round of review and you feel that this manuscript is now acceptable for publication, you may indicate that here to bypass the “Comments to the Author” section, enter your conflict of interest statement in the “Confidential to Editor” section, and submit your "Accept" recommendation.

Reviewer #2: All comments have been addressed

2. Is the manuscript technically sound, and do the data support the conclusions?

Reviewer #2: Yes

3. Has the statistical analysis been performed appropriately and rigorously? 

Reviewer #2: N/A

4. Have the authors made all data underlying the findings in their manuscript fully available?

Reviewer #2: Yes

5. Is the manuscript presented in an intelligible fashion and written in standard English?

Reviewer #2: Yes

6. Review Comments to the Author

Reviewer #2: The authors have revised the manuscript. My concerns have been addressed. I do not have further comments.

7. PLOS authors have the option to publish the peer review history of their article (what does this mean?). If published, this will include your full peer review and any attached files.

Reviewer #2: No

---

## [Editor Report · Acceptance letter]

7 Dec 2020

PONE-D-20-18250R1 

Prevalence and virulence gene profiles of *Escherichia coli* O157 from cattle slaughtered in Buea, Cameroon 

Dear Dr. Akomoneh:

I'm pleased to inform you that your manuscript has been deemed suitable for publication in PLOS ONE. Congratulations! Your manuscript is now with our production department. 

Kind regards, 

on behalf of

Dr. Iddya Karunasagar 

Academic Editor

PLOS ONE